# Amplified warming and marine heatwaves in the North Sea under a warming climate and their impacts

Bayoumy Mohamed<sup>1,2,\*</sup>, Alexander Barth<sup>1</sup>, Dimitry Van der Zande<sup>3</sup>, and Aida Alvera-Azcárate<sup>1</sup>

- <sup>1</sup>GeoHydrodynamics and Environment Research (GHER), University of Liège, Liège, Belgium
- <sup>2</sup>Oceanography Department, Faculty of Science, Alexandria University, Alexandria 21500, Egypt
  - <sup>3</sup> Operational Directorate Natural Environment, Royal Belgian Institute of Natural Sciences, Brussels, Belgium
  - \* Correspondence to: Bayoumy Mohamed (bayoumy.mohamed@uliege.be)

Abstract: The Northeast Atlantic and adjacent regions, such as the North Sea, are among the fastest-warming areas in the world. However, the role of climate change and internal variability on marine heatwaves (MHWs) in this region remains poorly understood. This study aims to quantify the relevant changes in sea surface temperature (SST) and MHWs in the North Sea, as well as to identify the leading patterns of interannual MHW variability over more than four decades (1982-2024). Our results indicate a new regime shift in the annual mean SST in the North Sea since 2013. Therefore, we examine the relationships between MHW trends and long-term SST warming trends to quantify the role of climate change in the intensification of MHWs. We found that the increase in MHWs is related to the significant decadal change in SST over the North Sea, and we have revealed that large-scale climate modes, such as the Atlantic Multidecadal Oscillation and the East Atlantic Pattern, play a crucial role in this decadal change in SST. In particular, the SST trend has doubled in the post-2013 period (0.8°C/decade) compared to the pre-2013 period (0.4°C/decade), leading to longer and more frequent MHWs. The SST, MHW frequency, and MHW days increased significantly by 0.38°C/decade, 1.04 events/decade, and 17.27 days/decade, respectively, over the entire study period. After removing the long-term SST warming trend before MHW detection, all MHW features exhibited insignificant trends, indicating that the long-term SST trend is the primary driver of the observed long-term MHW trend in the North Sea region, thereby confirming the crucial role of mean SST changes in MHW in this region. Furthermore, we found that 80% of the observed trend in MHW frequency is attributed to long-term warming, while the rest is attributed to internal variability. The SST record in May 2024, manifested by the longest (27 days) and most intense (2.2°C) MHW event, is attributed to an anomalous anticyclonic atmospheric circulation over the Baltic Sea and southern Norway, which enhances solar radiation over the North Sea. Finally, we also investigated how the chlorophyll-a concentration responded to the MHW, revealing a decrease in the deep and cold-water regions of the northern North Sea and an increase in the shallow and warm water areas of the southern North Sea.

### 1. Introduction

With anthropogenic global warming, extreme sea surface temperature (SST) events, such as marine heatwaves (MHWs) have increased worldwide, posing a challenge for scientific research and public policy (IPCC 2021). However, the increase in these events is neither temporally nor spatially uniform but varies depending on the period and geographical region under consideration. Quantifying these extreme events and assessing their key drivers and impacts has been one of the biggest challenges for climate research in recent years. The changes caused by climate change and the escalation of these extreme events are closely linked to human activities. These extreme events exacerbate thermal stress for marine organisms even more than shifts in mean climate conditions (Harris et al., 2018; Villeneuve and White, 2024). Researchers show that prolonged MHW events significantly impact aquatic ecosystems and fisheries and subsequently affect human life (Harris

et al., 2018; Villeneuve and White, 2024). MHWs have been documented in all the world's oceans, from the Arctic to the Antarctic (Pecuchet et al., 2025), and in marginal seas (Hamdeno et al., 2024; Mohamed et al., 2022; Pujol et al., 2022). They occur not only on the surface but also in the subsurface and can extend to the seafloor (Wilson et al., 2025), affecting marine ecosystems at different depths.





Recent studies have reported a significant increase in the frequency and duration of MHW in the Northeast Atlantic (Berthou et al., 2024; Chen and Staneva, 2024; Jacobs et al., 2024; Simon et al., 2023a), and in the North Sea (Chen et al., 2022; Mohamed et al., 2023). Generally, MHWs in these shallow water regions are more sensitive to regional and local atmospheric processes (e.g., increased solar radiation, lower winds and cloud cover, and tropical air). For the southern North Sea, the causal factors for the MHW have been linked to atmospheric circulation patterns and the interannual variability of climate modes, such as the Atlantic Multidecadal Oscillation (AMO), the North Atlantic Oscillation (NAO), and the East Atlantic Pattern (EAP). These large-scale climate modes are associated with SST and atmospheric variability in the North Atlantic, ranging from interannual (e.g., NAO) to decadal or longer timescales (e.g., AMO and EAP), which can influence the likelihood of MHW in this region (Holbrook et al., 2019). On the interannual time scale, the positive phases of the AMO and the EAP favor the development of a strong MHW, while the NAO makes the largest contribution only in winter (Mohamed et al., 2023). The presence and persistence of thermal stratification in the southern North Sea have been attributed to the occurrence of MHW, indicating the important role of MHW in the vertical structure of the water column (Chen et al., 2022).

In this study, we focus on the North Sea (Fig. 1A), a shallow, semi-enclosed northwestern European shelf sea with openings to the Atlantic Ocean in the north (Norwegian Sea) and in the south (English Channel), which is connected to the Baltic Sea in the east by the Skaggerak Strait. The North Sea is strongly influenced by the inflow of warm Atlantic water through the northern and southern openings, as well as less salty water from the Baltic Sea. The North Sea hosts large commercial fish populations and is considered one of the most productive fisheries in Europe, and a major marine ecosystem (Alvera-Azcárate et al., 2021; Ducrotoy et al., 2000). Climate-related changes and extreme events in this region could have a profound impact on this rich marine ecosystem (Kirby et al., 2007; Smale et al., 2019). These extreme events can also lead to shifts in species distribution, changes in biodiversity and community structure, and increased vulnerability to invasive species (Smale et al., 2019). MHW has also been found to contribute to oxygen depletion in the northern North Sea (Jacobs et al., 2024) and the Elbe estuary (Fan et al., 2025). Smale et al. (2019) identified the North Sea as an area where many species live near the edge of their thermal tolerance. MHWs in the North Sea in recent summers (2018-2022) have been associated with a collapse in dominant zooplankton populations, with physiological thermal limits exceeded for some species, indicating a significant impact of MHWs on zooplankton (Semmouri et al., 2023). MHWs are also likely to have an impact on chlorophyll-a concentration (CHL), which is a common indicator of phytoplankton biomass and essential for important biogeochemical processes (e.g., oceanic carbon sequestration and export). CHL in the North Sea is strongly influenced by sea surface temperature (SST), nutrient levels, and light conditions (Desmit et al., 2020). Recently, Alvera-Azcárate et al. (2021) pointed out the dominant role of SST on the timing of the spring bloom in the North Sea. They also observed a phenological shift, with the spring bloom occurring earlier each year, by about one month from 1998 to 2020. Generally, MHWs are associated with a decrease in CHL in the tropics and mid-latitudes, and an increase at high latitudes (Noh et al., 2022). However, the response of CHL to MHWs in the North Sea remains unclear.

In the northeast Atlantic and adjacent regions such as the Bay of Biscay, the Celtic Sea, the English Channel, and the North Sea, a regime shift occurred at the end of the last century (1990s) (Alheit et al., 2019; Biguino et al., 2023). This shift was caused by complex ocean-atmosphere interactions that led to large-scale changes in the strength and direction of the current systems that move the water masses in the North Atlantic and caused a decline in the Atlantic Meridional Overturning Circulation (AMOC) (Marzocchi et al., 2015). This climate shift was also accompanied by a significant weakening of the NAO (Robson et al., 2012) and a strong increase in the AMO (Biguino et al., 2023; Hughes et al., 2012). More than 25 years have passed since this regime shift (Fig. 1A) in the North Sea. Our research suggests that a new shift has recently occurred in the North Sea (Fig. 1B). A broad application of climate variability and prediction related to these regime shifts would be possible. For example, these are crucial for understanding the dynamics that amplify regional MHWs.

Our overarching goal in this study is to quantify the role of climate change and internal variability in the occurrence of MHW in the North Sea over more than four decades (1982–2024). Therefore, we focus on the following questions. (1) Has there been a recent new shift in the SST regime in the North Sea? (2) How does the change in mean SST explain the trends in MHW characteristics? (3) What are the main causes of the strongest and most intense MHW in spring 2024? (4) What are the anomalous responses of CHL to MHW during the overlap period with chlorophyll data (1998–2024)?

## 2. Materials and Methods

#### 2.1 Datasets


In this study, we used the daily NOAA Optimum Interpolation Sea Surface Temperature Dataset version 2.1 [OISST, (Huang et al., 2021; Reynolds et al., 2007)]. This dataset covers the period from January 1982 to December 2024, with a spatial resolution of 0.25 degrees in both latitude and longitude. In addition, the atmospheric variables were extracted from the ERA-5 reanalysis products of the European Center for Medium-Range Weather Forecasts [ECMWF, (Hersbach et al., 2020)]. These datasets cover the same period and spatial resolution as the SST. From this product, we have used the daily mean surface air temperature (SAT, hereafter), the u- and v-components of the wind speed (measured at 10 m height), the mean sea level pressure (MSLP), and the geopotential height at 500 hPa, as well as the net shortwave radiation. The ERA5 data are used to analyze the anomalous synoptic pattern and weather conditions during the longest and most extreme 2024 MHW event. Furthermore, we also used a monthly time series of climate indices representing the EAP and AMO climate modes. These climate mode data were obtained from the NOAA Physical Sciences Laboratory website (https://psl.noaa.gov/gcos\_wgsp/Timeseries/, accessed January 2025). Finally, we used the daily high-resolution (1 km) cloud-free chlorophyll-a concentrations from multi-satellites downloaded from the Atlantic Ocean Colour product (Copernicus-GlobColour Project, https://doi.org/10.48670/moi-00289, last accessed February 2025). We used this dataset to investigate the chlorophyll-a responses to MHW during the overlap period (1998–2024).

#### 2.2 Methods

100

All statistical analyses in this study were performed using the MATLAB program R2021a. For the detection and analysis of MHWs, the Marine Heatwaves Toolbox (Zhao and Marin, 2019), which implements the definition of Hobday et al. (2016), was applied to the daily SST data for each grid cell in the North Sea to detect the MHW characteristics. For trend analyses, seasonality, significance tests, Empirical Orthogonal Functions (EOFs), and graphical output, we mainly used the Climate Data Toolbox (Greene et al., 2019). A linear trend analysis was performed to determine the long-term trends of

SST, CHL, and MHW metrics (frequency, total number of days, and cumulative intensity). Then, the non-parametric Modified Mann–Kendall (MMK) test (Hamed and Ramachandra Rao, 1998; Wang et al., 2020) was used to determine whether the linear trends were significant at the 95% confidence level.

115

120

145

There are several well-documented methods for detecting regime shifts, which in modern climate change studies are defined as a rapid transition from one mean climate state to another (Rodionov, 2004; Zeileis et al., 2003). These methods are based on statistical hypothesis tests and can reproducibly identify regime shifts as significant abrupt changes in the SST time series. These methods can be used to test the significance of the occurrence of a single or multiple abrupt change points. Here, we employ the Pettitt homogeneity test (Pettitt, 1979), which is described in Biguino et al. (2023). The Pettitt test is a nonparametric test commonly used with hydrometeorological variables to determine the occurrence and timing of a single abrupt and significant change in the mean of a time series (Biguino et al., 2023). Then, we used the cumulative deviation test (Rebstock, 2002) to determine whether there are multiple change points in the SST anomaly (SSTA). To detect this regime shift or abrupt change points on the annual timescale, the seasonal cycle should be removed before the analysis (Pärn et al., 2021). Therefore, the daily SSTA was calculated by subtracting the long-term average SST for a given day (i.e., the daily climatology) from the observed SST of the same day; the monthly and annual SSTA were then calculated using the daily SSTA.

125 For the MHW calculations, we followed the methodology of Hobday et al. (2016), who define MHW as when the SST at a specific location is above the corresponding seasonally varying 90th percentile threshold for at least five consecutive days, based on a fixed reference baseline; here, we used the period 1982-2024 as the baseline climatology. In addition, we quantified the relative contributions of long-term warming (i.e., mean climate change) and internal variability to the observed trends in MHW characteristics. The long-term trends are most likely dominated by the external anthropogenic forcings, while the internal variability refers to natural fluctuations in the coupled ocean-atmosphere system, including 130 hydrodynamic processes (e.g., currents, mixing, stratification) and atmospheric variability (e.g., wind and pressure patterns). These processes can play a role in SST variability, which can amplify or suppress MHW development on seasonal to interannual time scales. For example, Mohamed et al. (2023) found that the change in atmospheric circulation over the southern North Sea in April 2013 led to an extremely cold event, while in the same month of the following year (2014) it 135 led to an extremely warm event. To isolate the influence of long-term SST warming, we first removed the long-term trend of SST at each grid point. Then, the MHWs were re-identified by calculating a new threshold using the detrended SST data and recalculating the corresponding MHW metrics (Lee et al., 2023). This approach refers to the "detrended baseline" (see Smith et al. (2025) for a discussion of the different baselines), which is also commonly referred to as one of the "shifting baselines" (Amaya et al., 2023).

Therefore, both a fixed and a linearly detrending baseline were used in our MHW calculations to isolate the effect of the long-term SST trend on the MHW metrics according to the following formula (Jin and Zhang, 2024; Lee et al., 2023; Simon et al., 2023b):

where MHW (SST) and MHW (SST detrended) are the MHW metrics derived from the original SST time series (i.e., the contribution of both long-term warming and internal variability to MHW) and the detrended SST time series (i.e., the

contribution of internal variability to MHW), respectively. The detrend method minimizes the impact of the increase in mean SST or long-term trend and analyzes the impact of changing SST variance. Moreover, we used the Trend Attribution Ratio (TAR) (Jin and Zhang, 2024; Li et al., 2023; Marin et al., 2021) to assess the relative contribution of the long-term SST trend and internal variability to the MHW trends:

TAR = 
$$\left(\frac{|rate^{trend}| - |rate^{detrend}|}{\max(|rate^{trend}|, |rate^{detrend}|)}\right)$$

where, |---| is the absolute value of the trends, and rate <sup>trend</sup> and rate <sup>detrend</sup> are MHW trends attributed to the long-term SST trend and the internal SST variability, respectively. The TAR value ranges from -1 to 1. If the TAR value is close to 1 (-1), the long-term SST trend (internal variability) is the dominant driver of the observed MHW trend. When the TAR value is close to or equal to 0, the SST's long-term trend and internal variability contribute equally to the observed MHW trends.

155 In addition to the default MHW parameters proposed by Hobday et al. (2016) (e.g., frequency, total days, and cumulative intensity), we calculated the probability ratio (PR) and fraction of attributable risk (FAR) for each MHW based on Frölicher et al. (2018). The PR can be interpreted as a measure of how the number of MHW days has changed each year relative to the total records. The PR is estimated as P1/P0, where P1 is the probability of MHW days in a specific year, defined as the total number of MHW days observed in that year divided by the total number of days in that year. P0 is the probability of 160 MHW days during the entire study period (1982–2024), defined as the number of MHW days observed in all years divided by the total number of days in all years (43 years\*365.25 days=15706 days). Thus, PR represents the relative strength of the MHW each year compared to the entire study period. If the PR is greater than 1, it indicates that the change in the MHW days in that particular year exceeds the local threshold throughout the entire record (the corresponding risk will be higher), and vice versa if the PR is less than 1. This risk factor is known as the fraction of attributable risk (Frölicher et al., 2018). The FAR value for each year is calculated from the PR value as FAR = 1 - (1/PR). The FAR values vary from 0 to 1 (or 165 0%-100%) if PR  $\geq 1$ . If PR  $\leq 1$ , which means that the change in the MHW days is less than the local threshold, the risk is zero. In addition to separating long-term warming and internal variability, we also analyze the anomalous synoptic patterns and weather conditions during the longest and most intense MHW event in spring 2024 as a case study. The atmospheric parameter anomalies and the CHL anomaly (CHLA) were estimated in the same way as the SSTA.

170 For the CHL analysis, we first analyze the seasonal variation and spatial trend of CHL over the period (1998–2024). Then, to investigate the potential impact of MHWs and marine cold spells (MCSs) on the CHL concentration in the North Sea, we redetermined the characteristics of MHWs and MCSs based on the climatological baseline of the period overlapping with the CHL (1998–2024). Subsequently, the CHLA is correlated with the total number of MHW days and MCS days.

#### 3. Results and Discussion

180

### 3.1 Regime shift and accelerated warming of the SST over the North Sea

To identify significant SST decadal changes, we analyzed the mean annual time series of SST over the North Sea. The horizontal distribution of the Pettitt test results is shown in Figure 1A. This non-parametric test allows us to determine the abrupt changing point in the SST time series that caused their heterogeneity (Biguino et al., 2023). The result of this test, based on the entire study period (1982–2024), reveals that a significant change point in SST occurred between 1996 and 2001 (Fig. 1A, top left panel). This period coincides with the AMO inversion from the negative to the positive phases

(Mohamed and Skliris, 2025). This climate shift took place mainly over the southern and eastern parts of the North Sea in 1997 and over the northwestern part of the North Sea in 2001. This result is consistent with that found on the Iberian coast and in the Northeast Atlantic (Alheit et al., 2019; Biguino et al., 2023). As this regime shift is well documented for various subregions of the Northeast Atlantic, including the North Sea (Alheit et al., 2019), we then reapplied the Pettitt test for the second period following this shift (2001 to 2024) to investigate whether a new regime shift has recently occurred in the North Sea SST. Figure 1A (top right panel) shows that a significant new abrupt change in SST occurred mainly in 2013 over most of the North Sea, except for the northwestern part, where this shift occurred in 2009. The annual mean SST time series indicates this regime shift with a dominant positive SSTA after 2013. The annual SST has increased significantly by around 0.8°C in recent years (2013–2024: post-2013, hereafter) compared to the previous period (1982–2012: pre-2013, hereafter), with an average SST of 10.67°C and 11.46°C in the pre- and post-2013 periods, respectively. The second climate shift is associated with strong positive phases of the EAP during the post-2013 period.

For additional validation, the cumulative deviation test (Rebstock, 2002) is applied to corroborate a significant regime shift in the SST over the North Sea. The annual cumulative SSTA time series shows a prevalence of negative anomalies (i.e. decreasing tendency) until the first climate shift between 1996 and 2001, followed by a slight increase until 2013, while a prevalence of positive anomalies (i.e. increasing tendency) is observed in the post-2013 period (Fig. 1B). These changing tendencies indicate a significant climatic shift after 2013, with an increase in the mean and trend of SST compared to the pre-2013 period. The observed SST warming trend appears to be pronounced post-2013, with an estimated rate of 0.8°C/decade, compared to 0.4°C/decade during the pre-2013 period.

The regional mean of the daily SSTA time series was calculated by averaging the daily SSTA data of all grid cells in the North Sea throughout the entire study period (1982–2024) and plotted using a Hovmöller diagram (daily vs. annual), as shown in Figure 1C. There was a strong temporal evolution of the average SSTA, dividing our study period into three distinct periods. The cold period (1982–2000), in which negative SSTA and MCS are predominant. This was followed by a transition period (2001–2012), in which both positive/negative SSTA and MHW/MCS can be observed. In the period after 2013, the North Sea warmed dramatically and transitioned to a warmer state, with a strong increase in SSTA and MHW (Fig. 1C and Fig. 2).

Figure 2 shows the evolution of the monthly SSTA over time during the entire study period (1982–2024). The SSTA ranged from -1.8 to 1.6°C above the seasonal climatological baseline (1982–2024), with the lowest value recorded in 1986 and the highest in May 2024 (yellow star in Fig. 2, which will be explained in more detail in the following sections). Considering the approach of using monthly data instead of daily data to determine the MHW based on the fixed threshold (Capotondi et al., 2024), there are 52 warm events (yellow circles in Fig. 2) where the SSTA exceeds the fixed 90<sup>th</sup> percentile threshold (red dashed line in Fig. 2) and 50 cold events (green circles) where the SSTA is below the fixed 10<sup>th</sup> percentile threshold (blue dashed line). There is evidence of an increased frequency of warm events, with 32 warm events occurring in the post-2013 period, compared to 20 warm events in the pre-2013 period. Most of the MCS events occurred mainly before the first regime shift. The monthly mean SSTA trend was estimated using the locally weighted scatterplot smoothing method (LOWESS; Cheng et al., 2022) and showed an increasing rate of 0.38°C/decade between 1982 and 2024. The LOWESS trendline (thick black line in Fig. 2) also confirms the acceleration of SST warming after the crucial point of climate shift in the post-2013 period. These results indicate that SST in the North Sea has undergone two significant regime shifts: the

first occurred between 1996 and 2001, the second after 2013. Therefore, in the next section, we focus on the most recent climate shift (post-2013) to investigate its role in amplifying MHW occurrence in the North Sea.

Figure 1: Regime shift of SST in the North Sea: (A) the horizontal distribution of significant change points (years) of the SST time series based on the Pettitt test during the whole period (1982–2024, top left) and in the last period (2001–2024, top right). (B) Long-term variation in cumulative SST anomalies throughout the entire period. The vertical yellow shading and the red line represent the corresponding years with the first abrupt SST changes (between 1996 and 2001) and the second in 2013, respectively. (C) Hovmöller diagram (daily vs. annual) of spatially averaged daily SST anomalies (seasonal cycle removed) between 1982 and 2024. The main geographical features and the isobaths (grey contour lines) of 20, 40, and 200 m are shown in panel (A).

Figure 2: Temporal evolution of regionally averaged sea surface temperature anomalies (SSTA) between 1982 and 2024. The thick black line represents the SSTA trend calculated with the locally weighted scatterplot smoothing method (LOWESS). The yellow/green circles show warm/cold events where SSTA was greater/smaller than the defined 90/10 percentiles (red/blue dashed horizontal lines), which were chosen to be seasonally independent for simplicity (i.e., fixed thresholds). The yellow star refers to the highest SSTA value recorded in May 2024.

#### 3.2 Pre-2013 versus post-2013 Marine Heatwaves

To investigate the regions with significant changes in mean SST, we examined the spatial distribution of mean SST differences between the post- and pre-2013 periods. The composite SST difference showed significant warming anomalies with average anomalies of 0.77°C over the entire North Sea, exhibiting an increasing zonal SST gradient from west to east (Fig. 3A). The largest anomalies were observed along the Belgian, Dutch, and Danish coasts, with an increase of up to 1.5°C in the German Bight and the Kattegat region. As highlighted in a previous study (Mohamed et al., 2023), a higher SST trend and MHWs were observed in the German Bight. These results suggest that the North Sea experienced a significant increase in mean SST after 2013, which could increase the probability of MHWs (Frölicher and Laufkötter, 2018). Therefore, we followed Frölicher et al. (2018) to investigate how the increase in mean SST could affect MHWs. Specifically, we quantified the annual mean probability ratio (PR, i.e. the fraction by which the number of MHW days changed per year) and the FAR of MHW days, as well as the relative change in the annual mean of the MHW spatial extent (i.e. the average area of a single MHW) (Fig. 3B). The regional changes in the PR and FAR of the MHW days were also calculated between the two periods (post-and pre-2013) divided by the pre-2013 period (Fig. 3C-D).

During the period before the first climate shift (i.e., from 1982 to 2001), the PR was less than one, and the FAR was almost zero (Fig. 3 B), which means that the likelihood of occurrence of the MHW days is very low compared to the entire study period. During the transition period, only 2002/2003 and 2006/2007 had a higher PR and FAR. During the post-2013 period, the PR increased by 2-5 times and the FAR also shows an increase between 25 % and 80 % (Fig. 3B). Throughout the entire study period, the most active MHW years (2006, 2007, 2014, 2020, 2022, 2023, and 2024), were those that showed an increase in FAR (more than 60 %). An increase in the spatial extent of the MHW was also observed in these years (black line in Fig. 3 B). The observed temporal development of the annual mean SST (green line in Fig. 3B) correlates strongly with the PR (r = 0.80, p 

Figure 3: (A) Composite differences in annual mean sea surface temperature between 2013–2024 and 1982–2012. (B) Annual mean values of the probability ratio (PR, blue bars), the fraction of attributable risk (FAR, red bars) of MHW days, the spatial extent of MHW (black line), and the sea surface temperature anomaly (SSTA, green line) throughout the entire study period. (C and D) The regional changes in the PR and FAR of MHW days in the post-2013 period compared to the pre-2013 period. Based on the standard two-sample Student's t-test, the composite mean difference is significant (p < 0.05) over the entire North Sea at a 95% confidence interval.

To explore how an increase in mean SST might affect the occurrence of MHW on a daily time scale, we analyzed the probability distribution function (PDF) of the daily area-averaged SSTA and the frequency of MHW occurrences in the North Sea, as shown in Figure 4. The PDF is estimated based on the Generalized Extreme Value (GEV) distribution (Li et al., 2023). Generally, the increase in internal variability of SST leads to a broadening of the PDF of temperature, making the occurrence of MHW more likely. In addition, changes in mean SST values due to the SST warming shift the center of the PDF to higher values, which also leads to an increased occurrence of MHW (Xu et al., 2022). To verify this in our study region, we compared the change in the PDF of daily SSTA between the pre-and post-2013 periods (i.e., the last climate shift), based on the original (Fig. 4A) and detrended SSTA time series (Fig. 4B). The PDF of the original SSTA exhibits a significant peak shift in the mean SST value between the two different periods (Fig. 4A). For the post-2013 period, the peak density of the GEV-fit PDF reaches 63% at an SSTA of 0.57°C, compared to 52% at an SSTA of -0.22°C in the pre-2013 period, with a statistically significant difference of 0.79 °C between the two periods. During the post-2013 period, there has been a clear shift towards a warmer state, indicated by the substantial increase in SSTA at the right-hand tail of the PDF, leading to an increase in the MHW occurrence, as shown by the area to the right of the green line marking the 90<sup>th</sup> threshold (Fig. 4A). Meanwhile, the SSTA variance changes slightly (from 0.56 °C to 0.39 °C). The curve shape of the post-2013 period is notably skewed (with a skewness value of -0.65), compared to a value of 0.02 in the pre-2013 period (Fig. 4A). Consequently, the occurrence of MHWs on the skewed side would be more frequent and intense than MCSs on the nonskewed side.








To further investigate the MHW occurrences between the two periods, we calculated the frequency of MHW occurrences for each month based on the original (Fig. 4C) and detrended SST data (Fig. 4D). The comparison of the monthly MHW frequency between the pre- and post-2013 periods (blue and red bars in Fig. 4c) reveals a clear seasonal asymmetry. While most months post-2013 show an increase in MHW frequency, the most pronounced increase is observed from June to December, where the MHW frequency almost doubled (i.e., increases from an average of 1 event in the pre-2013 period to 2 events in the post-2013 period). This suggests that climate warming has a strong impact on MHWs in the second half of the year, which has also led to increased summer stratification and reduced vertical mixing in recent decades (Chen et al., 2022; Chen and Staneva, 2024). In contrast, the changes in MHW frequency in the winter and early spring months (February and March) are less pronounced, indicating a weaker influence of warming during this period. The monthly averages of the mean MHW intensity for the two periods (pre- and post-2013) show a clear annual cycle (green and black lines in Fig. 4C), with maximum values of 2°C in summer and minimum values of 1°C in winter. The mean MHW intensity in the post-2013 period showed a seasonal shift compared to the pre-2013 period, with a more intense (warmer) and longer summer season. These results suggest that severe MHWs have occurred more frequently over the North Sea since 2013, particularly with a marked increase in MHW during the summer and fall seasons. After using the detrended SSTA for the PDF calculations (Fig. 4B), we found that the mean SST for the post-2013 period was not statistically different from that for the pre-2013 period, while the variance decreased slightly. In addition, the occurrence of MHW decreased in the post-2013 period compared to the pre-2013 period (Fig. 4D). These results indicate that the increase in mean SST (or long-term warming), rather than SST variability, plays a dominant role in the development of the long-term trend in MHW over the North Sea.

Figure 4: Probability distribution function (PDF, lines) and histogram (bars) of the regional averaged daily sea surface temperature anomaly (SSTA) during the pre-2013 period (1982–2012, blue) and the post-2013 period (2013–2024, red), based on the original (A) and detrended SST data (B). The vertical green dashed line refers to the 90th threshold temperature of the MHW. Monthly mean MHW frequency (bars) and MHW intensity (lines) from the original (C) and detrended SST data (D) for the pre-2013 period (blue bars and green lines) and post-2013 (red bars and black lines).

#### 3.3 Effects of the SST long-term trend and variability on MHWs




In this section, we analyze how the interannual variability of SST and long-term warming affect the long-term MHW trend in the North Sea over the entire period (1982–2024). The sensitivity of MHW in response to an increase in mean SST varies regionally (Lee et al., 2023). To date, no study has evaluated the relative role of the long-term trend and internal variability on the MHW in the North Sea. Therefore, we first investigated the linear trend maps for the annual mean of SSTA and the variance over the full study period (Fig. 5). The SSTA trend is significant throughout the region, with an increasing zonal gradient from west to east (Fig. 5A). The strongest SST trends are observed over the Kattegat and the German Bight. The SSTA variance shows an insignificant trend across the North Sea, except for the German Bight and the Norwegian Trench, where significant negative variance trends are observed in these regions (Fig. 5B). Then, to investigate the spatiotemporal patterns of MHW and their linear trends in the North Sea, we perform an empirical orthogonal analysis (EOF) and the corresponding principal components (PC) for the MHW that is detected based on the original SST (i.e., without removing the trend). Since the cumulative intensity can simultaneously reflect the frequency, duration, and mean intensity of the MHW (Jin and Zhang, 2024), we applied the EOF analysis to the annual cumulative intensity of the MHW (Fig. 6).

The first two EOF modes of cumulative MHW intensity account for 78.9% of the total variance. The spatial pattern of the leading first EOF mode (EOF1, explains 69.5%) shows high positive values over the entire study area (Fig. 6A), indicating an in-phase increase in MHW over the entire North Sea. The highest variability in cumulative MHW intensity is found over the shallow central region of the study area and the Dogger Bank (see the location in Fig. 1A). These regions are influenced by both warm Atlantic Water from the north/south of the North Sea opening and cold water from the Baltic Sea. The corresponding principal component of EOF1 (PC1) shows a significant (at 95% using the MMK test) increasing linear trend, especially after 2001 (i.e. the first climate shift) and more pronounced after 2013 (i.e. the second climate shift), indicating that MHW in the North Sea was more intense and prolonged after these climate shifts (Fig. 6C). Moreover, PC1 shows significant interannual variability, with the highest substantial increase of the cumulative MHW intensity occurring in 2003, 2006/2007, 2014, and the last three years (2022–2024), which coincide with the positive phases of the AMO index (Fig. 6C). The strong 2003, 2014, and 2023 MHWs in the southern North Sea have been reported by (Berthou et al., 2024; Mohamed et al., 2023).







The second EOF mode (EOF2, explains 9.4 %) is a regional variability mode and shows a dipolar oscillation with opposite fluctuations between the northeastern part of the study area and the southern and northwestern parts (Fig. 6B). The maximum positive variability is found over the regions influenced by the inflow of the less salty and cold water from the Baltic Sea via the Skagerrak Strait. In contrast, the maximum negative variability is observed over the regions influenced by the inflow of warm and salty Atlantic water. The PC2 does not show long-term warming; it is therefore mainly projected onto the interannual and decadal variability (Fig. 6C). From the PC2 time series, an increase in the amplitude of oscillations is evident, especially when comparing the period before and after 2001. Therefore, PC1 is associated with the long-term trend, and PC2 with the variability, which may also contain variability and a trend. We also examined the relationship between the PCs and the normalized indices of AMO and EAP (bars and green line in Fig. 6C). The first PC1 showed a significant correlation (r) with both AMO (r = 0.66, p

Figure 5: Linear trend maps for the annual mean of (A) the sea surface temperature anomaly (SSTA) and (B) the variance of the SSTA during the period 1982–2024. The dotted areas indicate that the trend is insignificant at a 95% significance level.

Figure 6: The two leading EOF modes of cumulative MHW intensity in the North Sea (A and B), and their corresponding normalized principal components (C) over the period 1982–2024. In panel (C), the bars represent the normalized AMO index, and the green line is the EAP index. Note that the EOF analysis is based on the original SST (i.e., without removing the SST trend).

Quantifying the role of climate change on increasing MHW characteristics depends on identifying the long-term externally forced SST warming trend separately from changes in internal variability (Oliver, 2019; Xu et al., 2022). Therefore, we evaluate the influence of the long-term SST warming on the MHW by removing the linear trend from the original SST at each grid point to obtain a detrended SST. We then used the detrended daily SST to re-determine the MHW characteristics based on the same climatological baseline (1982–2024). Although the temporal evolution of SST in the North Sea shows signs of nonlinearity, particularly due to the two observed climate shifts and the notable increase in MHWs after 2013, we opted for the linear trend method because the nonlinear trend could remove important low-frequency natural variability along with the anthropogenic warming signal (Smith et al., 2025), as shown in the North Atlantic after 2000 (Huang et al., 2024). Linear trend removal was chosen to provide a baseline quantification of long-term changes over the entire study period, a method commonly used in previous studies (Jin and Zhang, 2024; Lee et al., 2023; Marin et al., 2021), to allow comparison with other studies and regions.

The relative contributions of the long-term trend and the internal variability of the SST to the overall increase of the MHW characteristics (frequency, total number of days, and cumulative intensity) from 1982 to 2024 are estimated and compared in Figure 7. The MHW frequency and MHW days calculated from the original SST (red lines in Figs. 7A and 7B) show significant upward trends of  $1.04 \pm 0.29$  events/decade and  $17.27 \pm 5.91$  days/decade, respectively. Over the entire study period (1982–2024), 80 MHW events and 1121 MHW days were recorded, with 55% of these events (44 events) and 60% of the days (677 days) occurring after the recent climate shift (i.e., post-2013 period). Notably, 26 events and 541 days were concentrated in five particularly active MHW years: 2014, 2020, 2022, 2023, and 2024. In contrast, the MHW frequency and MHW days obtained from the detrended SST (black lines in Figs. 7A and 7B) exhibit insignificant negative trends, leading to a decrease in the MHW frequency (from 44 to 15 events) and total number of MHW days (from 677 to 168 days) in the post-2013 period. The difference between the values derived from the original and the detrended SST (the former minus the latter, green lines in Figs. 7A and 7B) highlights the dominant influence of the long-term SST trend in the development of the more frequent and longer-lasting MHWs observed in the North Sea.

The time series of the cumulative MHW intensity from the original SST, the detrended SST, and their difference in the period from 1982 to 2024 are shown in Figure 7C. The trends of the MHW cumulative intensity of the original SST show a significant increase of  $4.23 \pm 1.98$  (°C. days)/decade, with the highest cumulative intensity in 2022 and 2023. The difference between the results from the original minus the detrended data shows a higher significant trend of  $5.24 \pm 1.55$  (°C. days)/decade (Fig. 7C). Compared to the original time series (red lines in Fig. 7), the MHW frequency, days, and cumulative intensity resulting from the detrended SST data (black lines in Fig. 7) show insignificant decreasing trends. These results indicate that the long-term SST trend is the main driver for the observed long-term trend of MHW in the North Sea region, which confirms the crucial role of the mean SST changes on the MHW in this region.

Figure 7: Temporal evolution of the annual (A) MHW frequency, (B) MHW days, and (C) MHW cumulative intensity from 1982 to 2024. The red and black lines in (A, B, and C) represent the time series calculated from the original and detrended SST, respectively. The green lines are the differences between the original and detrended results.





The spatial distributions of the decadal trends of the MHW characteristics over the entire study period (1982-2024) are shown in Figure 8. Both the MHW frequency and the MHW days show a statistically significant (p < 0.05) upward trend over the whole North Sea (Figs. 8A and B), while the MHW cumulative intensity fluctuated from an insignificant (p > 0.05) negative trend in the Norwegian Trench and a significant positive trend in the southern North Sea (Fig. 8C). The trend in MHW frequency ranged from 0.5 to 1.5 events/decade (Fig. 8A). The highest decadal trend in MHW frequency (up to 1.5 events/decade) is found in the English Channel, west of the Dogger Bank and in the central part of the southern North Sea. The decadal trend of MHW days varied between 6 and 26 days/decade, with the highest trends (up to 26 days/decade) over the central part of the southern North Sea (Fig. 8B). These trend patterns of MHW frequency and days are consistent with those of (Chen and Staneva, 2024), although they used a shorter period (1993-2022). An insignificant trend in the MHW cumulative intensity is observed over most of the North Sea, except for the south of the North Sea, the English Channel and around the Shetland Islands in the northwestern part of the study region, which show a significant positive trend (Fig. 8C). This could be due to the internal variability processes, such as the inflow of cold and less salty water from the Baltic Sea, which can affect the stability of the water column and lead to a local decrease in the mean MHW intensity, while in the southern North Sea region the intensity of the MHW increases due to various oceanic processes and heat transport mechanisms caused by the inflow of warm Atlantic water (Chen and Staneva, 2024). In general, the trend of MHW characteristics in the North Sea exhibits a high spatial variability, with the highest trend in the southern North Sea and the English Channel and the lowest trend in the northern North Sea and the Norwegian Trench.

Figure 8: Decadal trends of (A) MHW frequency (events/decade), (B) MHW days (days/decade), and (C) MHW cumulative intensity (°C. days/decade) between 1992 and 2024. The dotted areas in (C) indicate that the trend is insignificant at a 95% significance level.



To further investigate the drivers of MHW trends at the regional scale, we estimate the trend attribution ratio (TAR) for MHW frequency and MHW cumulative intensity (Fig. 9). The TAR metric assigns the proportion of the MHW trend over 43 years (1982–2024) that is attributable to long-term changes in mean SST and residual internal variability (Marin et al., 2021). The positive TAR values for both MHW frequency and MHW cumulative intensity over the entire North Sea (Fig. 9) indicate that the trends in MHW frequency and MHW cumulative intensity are largely explained by the long-term changes in mean SST. The TAR values ranged from 0.4 to 1 for MHW frequency (Fig. 9A), and from 0.3 to 1 for MHW cumulative intensity (Fig. 9B), with an average of 0.80 and 0.78, respectively. The lowest TAR values for the MHW cumulative intensity coincide with the same regions (i.e., German Bight, Norwegian Trench) that showed significant negative trends in SST variance (Fig. 5B). The possible explanation is that the changes in internal variability due to the Baltic Sea inflow, which can influence the stability of the water column, contribute to a decrease in the trend of MHW intensity in these regions (Chen and Staneva, 2024). These results imply that, on average, 80% (78%) of the observed trends in MHW frequency (cumulative intensity) are due to the long-term SST trend, while the remainder is attributed to internal variability.

Figure 9: Trend Attributional Ratio (TAR) maps of the trends in (A) MHW frequency and (B) MHW cumulative intensity over the period (1982–2024). The positive TAR values over the entire area show the stronger influence of the mean SST changes on the observed MHW trends in the North Sea.

#### 3.4 Spring 2024 New Record SST and MHW




In this section, we explore the MHW events that occurred in 2024 and the factors that contributed to the spring 2024 MHW event as a case study. As shown in Section 3.1, the highest monthly SSTA record in the North Sea was reached in May 2024 (yellow star in Fig. 2), this record was manifested by the extreme MHW from May 10 to June 5, 2024 (yellow shaded area in Fig. 10A). Here, we focused on the causes of this extreme MHW event. Daily SSTA remained exceptionally high, with May 2024 being the second warmest period on record for this time of year after July 2014 (Fig. 10B). The third SST record was observed in June and September 2023, which were associated with an exceptional MHW in northwestern Europe (Berthou et al., 2024). The most active MHW years during the entire study period were observed post-2013 climate shift, which demonstrated a strong positive SSTA (Fig. 3B) and at least five MHW events per year (Fig. 7). According to the MHW severity classification (Hobday et al., 2018), the intensity of all these events can be categorized as moderate MHW, where the SSTA exceeds the 90th percentile threshold anomalies (Fig. 10B).

The highest daily SSTA in 2024 was recorded in the first half of this year, mainly from mid-February to early June (red bars at the bottom of Fig. 10A). During this period, five MHW events were detected where the daily SST (red line in Fig. 10B) exceeded the 90<sup>th</sup> percentile threshold (dashed black line in Fig.10B). The duration of these events ranged from 5 to 27 days. The longest (27 days) and most intense (2.2°C) MHW event occurred between May 10 and June 5, 2024 (spring 2024 MHW, hereafter). This event covered most of the North Sea (96% of the total area), with the SSTA ranging from 0.55 to 4°C (Fig. 11A). The spatial pattern of the SSTA shows a zonal gradient with the lowest value in the west (around the coasts of the United Kingdom and the English Channel) and the highest value in the east (around the coasts of Norway and Denmark and in the German Bight). Atmospheric conditions during this event showed warm surface atmospheric temperature anomalies (SATA) over the entire region (Fig. 11B), with the same pattern as the SSTA over the ocean. The highest SATA values were found over the eastern coasts, which coincides with the same region experiencing the highest

MHW intensity. Over the land, the highest SATA was found in Norway and the lowest in France. This suggests that atmospheric overheating (i.e., the increase in air temperatures compared to their climatological values at this time of year) and thus weather conditions could best explain this MHW event.

Figure 10: Regionally daily averaged (A) sea surface temperature (SST) and (B) sea surface temperature anomaly (SSTA) from 1982 to 2024, represented by grey lines, except for the active MHW years (2014, 2020, 2022, 2023, and 2024). In (A), the dashed black/blue lines represent the climatology/90<sup>th</sup> percentile of daily SST. The daily SSTA during 2024 is shown in red bars at the bottom of panel (A), and the vertical yellow shading represents the MHW occurring from May 10 to June 5, 2024. In (B), the 90<sup>th</sup>/10<sup>th</sup> percentile anomalies are represented by black dotted lines. Note that all the anomalies were calculated by subtracting the climatological daily SST (computed over 1982–2024).

The question, therefore, arises as to what the main causes of this atmospheric overheating and the MHW event in spring 2024 are. We address this question by examining the decomposition of all atmospheric factors, including cloud cover, heat budget components, wind components, atmospheric pressure, and geopotential height. During this event, the total cloud cover anomalies over most of the North Sea are always negative, with the highest negative anomalies (up to 20%) occurring

in the core region with the highest intensity of the MHW (Fig. 11C). As a result of the reduction in total cloud cover, the shortwave radiation has increased over most of the North Sea (Fig. 11D), except in the region around the Strait of Dover where negative shortwave radiation anomalies are accompanied by positive cloud cover anomaly. The contrasting pattern of cloud cover and shortwave radiation demonstrated their relatedness. The average increase in shortwave surface radiation during this MHW event was 6 W/m<sup>2</sup> compared to the daily climatological mean at this time of year (calculated over 1982– 2024). These abnormal climate conditions could be responsible for the high atmospheric and oceanic temperatures that eventually triggered the spring MHW. This decrease in wind speed could enhance the thermal stratification of the water column, inhibit vertical mixing, and thus reduce downward heat transfer and lead to heat accumulation in the surface layers. In addition, the geopotential height anomalies at 500 hPa during this MHW event show a strong anomalous anticyclonic circulation centered over the northern Baltic Sea (Fig. 11F), with the area under the high-pressure anomalies (up to +3.5 hPa) dominated by subsidence. The edge of this circulation extended into the northern part of the North Sea. This anticyclonic circulation favors the reduction of total cloud cover, leading to increased solar radiation and further warming of the upper ocean through air-sea interactions (i.e., conditions that are favorable for the occurrence of the MHW). In addition, cyclonic atmospheric patterns with lower air pressure anomalies (less than -1.5 hPa) are observed over France and Belgium. As a result of these two atmospheric systems, the wind anomalies over the North Sea were dominated by easterly and south-easterly winds (arrows in Fig. 10E), which led to the transport of warm continental air masses over the sea. These results suggest that, in addition to long-term warming, which has contributed to the increase in the MHW occurrence in recent decades (Fig. 7), natural variability and atmospheric circulation may also play a role in modulating (i.e., either amplifying or attenuating) MHW characteristics (Chen and Staneva, 2024; Mohamed et al., 2023). Particularly in shallow water regions such as the North Sea, where atmospheric circulations over the North Atlantic and European region, including the North Sea, are likely to influence SST variability due to the rapid response of shallow water seas to atmospheric forcing (Atkins et al., 2024).




Figure 11: Anomalies of SST and atmospheric conditions during the spring 2024 marine heat wave (May 10–June 5, 2024). (A) SSTA, (B) surface air temperature anomaly (SATA), (C) total cloud cover anomalies (TCCA), (D) shortwave radiation anomalies (SRA), (E) wind speed anomalies (WSA, shading is the magnitude and vectors are wind direction), (F) geopotential height anomalies (GPHA) at 500 hPa, the dashed contour lines represent the anomalies of the mean sea level pressure (hPa). All anomalies were calculated relative to the daily climatological baseline (1982–2024).

## 3.5 Biological Impact of MHW and MCS





To investigate the biological impacts of MHW and MCS in the North Sea, we first analyze the seasonal variation and spatial trend of CHL over the period (1998–2024). Then the CHLA is correlated with the total number of MHW days and MCS days. The climatological daily CHL seasonal cycle (Fig.12) shows that the highest variability of CHL occurs in April and May, which are characterized by the spring phytoplankton bloom (Amorim et al., 2024), with an average monthly CHL concentration of 2.2, 3.1, 3.0, and 2.2 mg/m³ for March, April, May, and June, respectively. The monthly trend analysis of the CHL shows a small but still significant positive trend in February and March. In contrast, a significant negative trend was observed in May, while no significant trends were observed in the other months, as the uncertainties were higher than the trend values (boxplots in Fig.12). A possible explanation for the contrasting CHL trends (i.e., positive in February and March and negative in May) is that the onset of the spring bloom in the North Sea has been observed to start earlier each year (Alvera-Azcárate et al., 2021).

The comparison of the CHL concentration between the pre- and post-2013 periods is shown in Figure 12 (green and red lines). In the post-2013 period, the highest peak CHL concentration occurred in April instead of May, as in the pre-2013 period, confirming the change in the timing of the spring bloom in the North Sea. During the post-2013 period, CHL concentrations increased in February and the first half of March compared to the pre-2013 period, coinciding with a decrease in the MHW intensity during these months (see the green and black lines in Fig. 4C). Meanwhile, CHL concentrations decreased in May and June relative to the pre-2013 period, aligning with an increase in MHW intensity during these months (see the green and black lines in Fig. 4C). This suggests that the CHL concentration in the North Sea could be affected by MHW on a seasonal scale.

A high spatial variability of the monthly CHLA trends can be observed in the North Sea (Fig. 13A). The strongest negative trends (between -0.4 and -0.8 mg/m³/decade) are found in the north-eastern corner of the study area (i.e., the regions around the Norway and Denmark), along the French coast, in the Wadden Sea and the deep part west of the Dogger Bank (Fig. 13A). In contrast, the most significant positive CHL trends (between +0.4 and +0.8 mg/m³/decade) are recorded along the south-east coast of the UK and in the south-central part of the study area, which is influenced by nutrient-rich freshwater inflows from the rivers Thames, Bure, Waveney, Wang, Blyth, and Deben along the southernmost coast of the UK. Positive trends are also observed along the Belgian coast and in the eastern German Bight, which is influenced by nutrient-rich freshwater inflows from rivers, such as the Scheldt and Elbe. To this end, the CHL concentration in the North Sea showed a high variability both in the mean value (grey shaded area in Fig. 12) and in the trend (Fig. 13A). This raises the question of whether the MHW influences CHL concentrations in the North Sea.







Figure 12: (A) The average CHL daily seasonal cycle over the entire North Sea for the whole period (1998–2024, black line), preand post-2013, green and red lines, respectively. The grey shaded area represents the  $\pm$  standard deviation. The box plots show the monthly CHL trend with the associated uncertainty.

To better assess the possible effects of MHW and MCS events on the CHL concentration in the North Sea, the CHLA is correlated with the total number of MHW and MCS days (Fig. 13 B and C). The largest and most significant negative correlations between MHW days and CHLA are found in deep- and cold-water regions (i.e., in the central and north-eastern sector of the North Sea, including the Norwegian Trench and the Skagerrak and Kattegat straits) and along the French coast (Fig. 13B). In contrast, positive significant correlations are found in the shallow and warm waters of the southern North Sea, especially along the south-eastern coast of the United Kingdom, in the German Bight and the region in between (Fig. 13B). The opposite pattern of correlation is observed between CHLA and MCS days (Fig. 13C). In most other areas, the correlations are insignificant (the dotted regions in Fig. 13 B and C). The possible explanation for the contrasting response of CHL concentration to MHW (i.e., the negative response in the deep- and cold-water regions and the positive response in the shallow and warm regions) is that MHWs in deep- and cold-water regions can increase stratification and reduce vertical mixing. This restricts the upward transport of nutrients from deeper layers, leading to nutrient depletion in the euphotic zone and thus to lower CHL concentrations. In the shallow and warm water regions of the southern North Sea, these areas are less stratified and can be more easily mixed by wind or tides. MHWs can enhance metabolic rates and stimulate phytoplankton growth when nutrients are still available, leading to an increase in CHL concentrations.

To illustrate how the CHL changes over time, we have plotted the monthly average CHLA in the Norwegian Trench (the area bounded by the red contour region in Fig. 13A) as an example (Fig. 13D). The result shows that the CHL concentration in the Norwegian Trench decreases with a trend of -0.31 mg/m3/decade between 1998 and 2024. The average CHL concentration decreased from 1.7 mg/m³ in the pre-2013 period to 1.1 mg/m³ in the post-2013 period. Positive CHLA dominated in the pre-2013 period, with 28 high CHL events (yellow circles in Fig. 13D), where CHLA exceeded the fixed 90th percentile threshold (the horizontal red dashed line in Fig. 13D). In the post-2013 period, negative CHLA was prevalent and was associated with low CHL events (green circles) where CHLA was below the fixed 10th percentile threshold (the horizontal blue dashed line). Generally, the response of CHL concentration to the MHW in the North Sea is not uniform and varies between the different subregions. The complex interaction of CHL with the MHW confirms that there are other factors influencing CHL concentrations in the North Sea in addition to the sea surface warming, such as light conditions and changes in nutrient inputs from the rivers (Desmit et al., 2020; Jacobs et al., 2024). Therefore, further research is needed to investigate the response of CHL to MHW and the other factors on a regional and seasonal scale (e.g., during the spring bloom), especially in areas that show a positive or negative response of CHL to MHW.

Figure 13: (A) Trend map of CHLA (mg/m³/decade) and the correlation maps between (B) MHW days and CHLA and (C) MCS days and CHLA over the period between 1998 and 2024. (D) The average monthly CHLA over the Norwegian Trench (the area bounded by the red contour in panel A) between 1998 and 2024. The thick black line represents the LOWESS trendline. The yellow/green circles show high/low CHL events where CHLA was greater/smaller than the defined 90/10 percentiles (red/blue dashed horizontal lines), which were chosen to be seasonally independent for simplicity (i.e., fixed thresholds). In (A, B, and C), the dotted areas indicate that the trend or correlation is insignificant.

#### 4. Summary and Conclusions







In this study, we considered different MHW approaches, including fixed, seasonally varying thresholds and detrended methods, to investigate the relative contribution of the long-term warming trend and internal variability to the observed trends in MHW characteristics in the North Sea. We also investigated the underlying factors contributing to the spring 2024 MHW event and the responses of chlorophyll to MHW during the period (1998–2024). The main results are summarized as follows:

Over the 43 years (1982–2024), a significant increase in SST (0.38°C/decade), MHW frequency (1.04 events/decade), total days (17.27 ± 5.91 days/decade), and cumulative intensity (4.23 ± 0.1.98°C. days/decade) was observed in the North Sea. Once detrended SST was used for MHW detection, all MHW characteristics showed insignificant trends, suggesting that long-term SST warming was the main cause of the observed trend increase in MHW characteristics. Long-term SST warming contributed to 80% of the observed trend in MHW occurrence. These results confirm that the long-term SST trend, rather than the trend in SST variability, dominates the change in MHW in the North Sea. Our results showed an abrupt change in mean SST in the post-2013 period. This climate shift was associated with a twofold increase in the SST trend (0.8°C/decade) compared to the trend in the pre-2013 period (0.4°C/decade). This climate shift also led to a 64% increase in MHW frequency and a 24% increase in MHW duration compared to the pre-2013 period. The strong relationship between the mean SST and the MHW indicates that the MHW occurrence in the North Sea will continue to increase under global warming.

A new SST record was set in the North Sea in May 2024. This SST record was associated with the strongest (2.2°C) and longest (27 days, from May 10 to June 5, 2024) MHW event. In addition to long-term warming, the manifestation of this MHW event was mainly associated with the observed anomalous anticyclonic circulation over the Baltic Sea and southern Norway, which reduced cloud cover and wind speed and increased shortwave solar radiation over the North Sea. Consequently, these processes led to heat accumulation near the sea surface, which increased the SST and caused this MHW event. Finally, we investigated the complex relationship between MHW and surface chlorophyll-a concentrations in the North Sea. The results showed a contrasting response of chlorophyll-a to MHW, with a decrease in the deep- and cold-water regions of the northern North Sea (e.g., in the Norwegian Trench and the Skagerrak Strait) and an increase in the shallow and warm water regions of the southern North Sea. Further research is needed to predict future MHW in the North Sea under different climate scenarios and to investigate the extreme compound events (e.g., marine, atmospheric heatwaves, and heavy rainfall) and their consequences and impacts in the North Sea.

**Data Availability Statement:** The original contributions presented in the study are included in the article; further inquiries can be directed to the corresponding author.

**Conflicts of Interest:** The authors declare that they have no conflicts of interest, except that the last co-author (Aida Alvera-Azcárate) is a member of the editorial board of Ocean Science.

**Author Contributions:** Conceptualization: BM, AB, DVDZ, and AAA. Methodology and visualization: BM, AAA, DVDZ, and AB. Writing –original draft preparation: BM. Review and editing: BM, AB, DVDZ, and AAA. Supervision: AAA. All authors have read and agreed to the published version of the manuscript.

**Funding:** This work was fully funded by the STEREO-IV (Support To Exploitation and Research in Earth Observation) program administered by BELSPO (Belgian Science Policy Office) through the North-Heat project (STEREO-IV BELSPO # project SR/00/404).

Acknowledgments: The authors would like to thank the organizations that provided the data used in this work, including the Copernicus Marine Environment Monitoring Service (CMEMS), the European Centre for Medium-Range Weather Forecasts (ECMWF), and the National Oceanic and Atmospheric Administration (NOAA). The authors would also like to thank the anonymous reviewers for their contributions to the development of this manuscript.

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
