# Peer review of "Amplified warming and marine heatwaves in the North Sea under a warming climate and their impacts"

_EGUsphere, 2025_

## Author Comment (AC1)

General comments

This paper investigates the long-term impact of climate warming on the occurrence of marine heatwaves in the North Sea—a topic well within the scope of Ocean Science. The manuscript is well-written and clearly structured. The authors address the main research question appropriately and present interesting results that contribute to the scientific community. Overall, I believe the study meets the journal's standards and could be published after moderate revision. Detailed comments and suggestions for improvement are provided below.

**A: The authors would like to thank the reviewer for his/her very careful reading, positive feedback, valuable comments, and suggestions that helped us improve the manuscript. And for finding that our manuscript is well-written, structured, and has clear objectives. We have addressed all the comments below.**

Specific comments
P2_L50:
The authors state that "the variability of MHW in the southern North Sea has been attributed to changes in stratification…". I believe this is a misinterpretation of Chen et al. (2022). That study does not attribute MHW variability to stratification; rather, it argues the opposite—that the presence and persistence of stratification in the southern North Sea are attributed to the occurrence of MHWs.

**A: Thank you for pointing this out. We acknowledge the misinterpretation of Chen et al. (2022). In the revised manuscript, we have rewritten the sentence as follows.**

- **The presence and persistence of thermal stratification in the southern North Sea have been attributed to the occurrence of MHW, indicating the important role of MHW in the vertical structure of the water column (Chen et al., 2022).**

P2_L71:
Throughout the Introduction, there is no mention of chlorophyll-a, yet the final sentence abruptly introduces it as a focus question regarding its response to MHWs. Even earlier in the paragraph, the stated goal of the study is to quantify the role of climate change, specifically increasing SST, in MHWs. This sudden shift lacks coherence. I recommend that the authors either omit this focus (along with Section 3.5) and reserve it for a future study (which may already form a complete narrative), or revise the introduction to systematically incorporate this aspect.

**A:** Thank you for your insightful comment and your suggestion to reserve this section for a future narrative study. In the revised version of the manuscript, we added more detail about the CHL-a studies in the North Sea and the role of the MHW on the CHL-a from previous studies in other regions. We have highlighted these in the introduction and the methodology as follows.

- Climate-related changes and extreme events in this region could have a profound impact on this rich marine ecosystem (Kirby et al., 2007; Smale et al., 2019). These extreme events can also lead to shifts in species distribution, changes in biodiversity and community structure, and increased vulnerability to invasive species (Smale et al., 2019). MHW has also been found to contribute to oxygen depletion in the northern North Sea (Jacobs et al., 2024) and the Elbe estuary (Fan et al., 2025). Smale et al. (2019) identified the North Sea as an area where many species live near the edge of their thermal tolerance. MHWs in the North Sea in recent summers (2018-2022) have been associated with a collapse in dominant zooplankton populations, with physiological thermal limits exceeded for some species, indicating a significant impact of MHWs on zooplankton (Semmouri et al., 2023). MHWs are also likely to have an impact on chlorophyll-a concentration (CHL), which is a common indicator of phytoplankton biomass and essential for important biogeochemical processes (e.g., oceanic carbon sequestration and export). CHL in the North Sea is strongly influenced by sea surface temperature (SST), nutrient levels, and light conditions (Desmit et al., 2020). Recently, Alvera-Azcárate et al. (2021) pointed out the dominant role of SST on the timing of the spring bloom in the North Sea. They also observed a phenological shift, with the spring bloom occurring earlier each year, by about one month from 1998 to 2020. Generally, MHWs are associated with a decrease in CHL in the tropics and mid-latitudes, and an increase at high latitudes (Noh et al., 2022). However, the response of CHL to MHWs in the North Sea remains unclear.

- In the methodology, we have added more details as follows: For the CHL analysis, we first analyze the seasonal variation and spatial trend of CHL over the period (1998–2024). Then, to investigate the potential impact of MHWs and MCSs on the CHL concentration in the North Sea, we redetermined the characteristics of MHWs and MCSs based on the climatological baseline of the period overlapping with the CHL (1998–2024). Subsequently, the CHLA is correlated with the total number of MHW days and MCS days.

P5_L155–160:
I would not describe 2012/13 as indicating a "second regime shift." A regime shift is not instantaneous; it marks a transition that may unfold over months or years. Figures 1a and 2 clearly illustrate such a transition. The regime shift appears to begin around 2000 and conclude in 2012. Rather than defining 2012/13 as a second regime shift, this study might more accurately be described as the first to delineate the full span of regime shift—from 2000 to 2012. Clarifying this timeframe provides greater scientific value than introducing an arguably redundant second shift.

**A: We fully agree with you that the regime shift is not instantaneous, and it could take a longer period. For this reason, we considered a robust statistical technique to detect a single (e.g., Pettitt test, Fig. 1A) or multiple (e.g., cumulative deviation test, Fig. 1B) abrupt change points of mean SST. In addition, we consider the period between the two shifts as a transition period, where the SSTA fluctuated between negative and positive anomalies. We also used the LOWESS regression (black line in Fig. 2), which confirms the second shift and the accelerated trend in the post-2013 period compared to the previous period.**

P5_L173–175:
According to the domain-averaged SSTA (the thick black line in Figure 2), 2000 seems to be the transition point between the cold and transitional periods, as it is when the averaged SSTA reaches 0°C. Moreover, Figure 1 shows that, based on the Pettitt test, 70% of the North Sea experienced the transition after 2000, while only 30% (mainly the southern North Sea) transitioned between 1996 and 1998. If 1997 is used as the transition year, the affected area would be less than 10%.

**A: Thank you for this useful comment. Although we emphasized that this transition took place between 1996 and 2001 (vertical yellow shading in Figure 1B), your suggestion is better to consider the transition after 2000, as it covers a larger area. Therefore, we have changed the sentence as follows.**

- **There was a strong temporal evolution of the average SSTA, dividing our study period into three distinct periods. The cold period (1982–2000), in which negative SSTA and a marine cold spell (MCS) are predominant. This was followed by a transition period (2001–2012), in which both positive/negative SSTA and MHW/MCS can be observed. In the period after 2013, the North Sea warmed dramatically and transitioned to a warmer state, with a strong increase in SSTA and MHW (Fig. 1C and Fig. 2).**

P6_L187–189:
The authors state that "SST in the North Sea experienced two significant regime shifts in the late 1990s and after 2013." In my view, the North Sea underwent a single regime shift between 2000 and 2012, transitioning from MCS dominance to MHW dominance.

**A: We identified two statistically significant shifts in SST based on changepoint detection methods: the first occurred between 1996 and 2001, the second after 2013 (Fig. 1B). However, we acknowledge that these shifts can also be interpreted as part of a broader, more gradual transition that occurred between 2001 and 2012, as you suggest. Therefore, we consider the period between 2001 and 2012 as a transition period between the two shifts. In this transition period, both positive/negative SSTA and MHW/MCS were observed, while in the post-2013 period, only positive SSTA and MHW were dominant.**

P7:
Following my previous comment, I suggest that the authors indicate the regime shift period (2000–2012) in Figure 2.

**A: In the revised version, we consider this period as the transition period between the two shifts.**

P9_L251–252:
The authors state: "The increase in internal variability of SST leads to a broadening of the PDF of temperature, making the occurrence of MHW more likely." However, Figure 4A–B shows a decrease in variance from the pre-2013 to post-2013 period. Does this imply that MHWs became less likely after 2013? This needs clarification.

**A: This sentence is a general conclusion from Xu et al. (2022). For this reason, we follow it with another clarifying sentence, "To verify this in our study region, we compared........." (please see lines 153).**

 P10_L266–268:
The authors write: "The frequency of MHW occurrence is higher in all months post-2013 than pre-2013, except for February and March…" In my view, a more outstanding difference is post-2013, the mean MHW frequency is considerably higher (almost doubled) than pre-2013 from June to December. This implies that climate warming mainly affects the appearance of MHW in the second half of the year. While this is mentioned in lines 272–273, the earlier description (lines 266–271) does not clearly highlight it.

**A: That's a very good point. In the revised version, we have highlighted this as follows.**

- **To further investigate the MHW occurrences between the two periods, we calculated the frequency of MHW occurrences for each month based on the original (Fig. 4C) and detrended SST data (Fig. 4D). The comparison of the monthly MHW frequency between the pre- and post-2013 periods (blue and red bars in Fig. 4c) reveals a clear seasonal asymmetry. While most months post-2013 show an increase in MHW frequency, the most pronounced increase is observed from June to December, where the MHW frequency almost doubled (i.e., increases from an average of 1 event in the pre-2013 period to 2 events in the post-2013 period). This suggests that climate warming has a strong impact on MHWs in the second half of the year, which has also led to increased summer stratification and reduced vertical mixing in recent decades (Chen et al, 2022, 2025). In contrast, the changes in MHW frequency in the winter and early spring months (February and March) are less pronounced, indicating a weaker influence of warming during this period.**

P11_L285–287:
The statement "To date, no study has evaluated the relative role of the long-term trend and internal variability on the MHW in the North Sea" is not true. Chen and Staneva (2024) have addressed this very question, using similar data (1982–2022) and methodology (Hobday et al., 2016; MATLAB toolbox by Zhao & Marin, 2019). They also identified different MHW patterns over the last 30 years (1993–2002, 2003–2012, 2013–2022). To my knowledge, their study is the first of its kind in the North Sea. The authors should revise this claim and properly credit prior research, especially work so closely aligned with theirs.

Chen, W., & Staneva, J. (2024): Characteristics and trends of marine heatwaves in the northwest European Shelf and the impacts on density stratification: In: von Schuckmann, K., Moreira, L., Grégoire, M., Marcos, M., Staneva, J., Brasseur, P., Garric, G., Lionello, P., Karstensen, J., and Neukermans, G. (eds.): 8th edition of the Copernicus Ocean State Report (OSR8). Copernicus Publications, State Planet, 4-osr8, 7, doi:10.5194/sp-4-osr8-7-2024

**A: Thank you for providing us with this very valuable study. In the revised version, we have included it with emphasis on their main findings. We would like to draw your attention to the fact that this study did not consider removing the SST trend before the MHW detection to evaluate the relative role of the long-term trend and internal variability.**

P14_L356:
Why were specific years selected rather than showing long-term trends? While Figure 8 provides spatial maps of annual mean MHW days, selecting individual years only highlights temporal variability within the same region (i.e., the North Sea). For instance, although 2022 and 2023 had similar total MHW days, the southern North Sea experienced different MHW durations. Presenting trends instead, like in Figure 2e–h of Chen and Staneva (2024), would better illustrate spatial variability.

**A: Thank you for this valuable suggestion. In the revised version, we have removed this figure and show the spatial trends instead. In this new figure, we calculate the trend of MHW frequency, total days, and cumulative intensity. And discuss these results with those of Chen and Staneva (2024).**

P14_L365–367:
The statement that "SST variability and thus MHW in the North Sea are largely influenced by atmospheric rather than oceanic forcing, which is consistent with Tinker and Howes (2020)" is misleading. Tinker and Howes (2020) found that marine air temperature is the main driver of SST rise—not necessarily the dominant influence on MHWs.

**A: Based on your above comment, we have removed this paragraph and replaced this figure and related text with the trend of the MHW frequency, total days, and cumulative intensity.**

P15_L379–381:
Why focus only on frequency and intensity? I would expect a discussion of trends in MHW duration and total days, or at least an exploration of the drivers behind MHW characteristics.

**A: In the revised version, we have added the temporal evaluation and trend of the total number of MHW days (Fig. 7B) as well as their spatial trend (Fig. 8B).**

P15_L387:
What exactly is meant by "internal variability"? Do the authors refer to hydrodynamics?

**A: Here, internal variability refers to the change in the SSTA Variance due to the Baltic Sea inflow. We have clarified this point as follows.**

**The possible explanation is that the changes in internal variability due to the Baltic Sea inflow, which can influence the stability of the water column, contribute to a decrease in the trend of MHW intensity in these regions (Chen and Staneva, 2024).**

P18_L440:
It's not only reduced wind mixing; stable stratification also suppresses vertical water mass exchange, thereby limiting heat transfer to deeper layers. As a result, heat accumulates in the surface layers.

**A: We have rewritten the sentence as follows:**

- **This decrease in wind speed could enhance the thermal stratification of the water column, inhibit vertical mixing, and thus reduce downward heat transfer and lead to heat accumulation in the surface layers.**

P19_Section 3.5:
As previously mentioned, I do not see a clear connection between this section and the overall focus of the paper (nor is it reflected in the title). Either develop the introduction and methodology to properly integrate this topic, or consider removing it and addressing it in a future study.

**A: Thank you for this insightful comment. In the revised version of the manuscript, we have added and highlighted more details about CHL in the introduction and methodology to improve the coherence of the manuscript and integrate this section to clarify its relevance to the manuscript theme.**

P22_Conclusions:

The conclusion section is overly long and verbose. Please revise to make it more concise and focused.

**A: We have revised the conclusion section to make it more concise and focused.**

Technical Corrections

P4_L125: Add a comma after the equation.

**A: done.**

P4_L126: Change "Where" to lowercase: "where".

**A: done**

**We hope these revisions and clarifications address your concerns. Thank you once again for your valuable feedback and constructive comments.**

---

## Author Comment (AC2)

**Review of "Amplified Warming and Marine Heatwaves in the North Sea Under a Warming Climate" by Bayoumy Mohamed et al**

**Overview**

The manuscript uses a long-term dataset of sea surface temperature (SST) observations in the North Sea to study the impact of the warming climate on SSTs and marine heatwaves (MHWs). The authors use established statistical techniques to confirm a regime shift in the 1990s, identify a further change in 2013, and attribute much of the recent increase in MHW occurrence and severity to increases in SST. MHWs strongly impact the marine environment, particularly the ecosystem and this study, showing ocean warming to be a major driver, is important for understanding how MHWs may develop in the future.

In addition to the SST/MHW analysis, a case study of a strong MHW event in spring 2024 and the impact of hot and cold events on chlorophyll-a concentrations are also studied.

**A: The authors would like to thank the reviewer for his/her time and effort in providing valuable comments and suggestions that helped us improve the manuscript. And for the very careful reading of the manuscript and the positive feedback. We have addressed all the comments below.**

**Main comments**

The analysis of the spring 2024 event and chl-a concentrations are rather disconnected from the SST/MHW analysis (sections 3.1 – 3.3) and not related to the manuscript title. These sections should be better integrated in the manuscript. For instance, the 2024 event is attributed to anomalous atmospheric conditions; would the event still have happened without SST warming? Also, the description of changes in the timing of the chl-a cycle pre- and post-2009 is interesting but difficult to relate to the SST or MHW changes which are shown for different time periods (pre- and post-2013).

**A: Thank you for this insightful comment. In the revised version of the manuscript, we have added and highlighted the following to improve the coherence of the manuscript and integrate these sections to clarify their relevance to the manuscript theme.**

**In the introduction:**

- **Climate-related changes and extreme events in this region could have a profound impact on this rich marine ecosystem (Kirby et al., 2007; Smale et al., 2019). These extreme events can also lead to shifts in species distribution, changes in biodiversity and community structure, and increased vulnerability to invasive species (Smale et al., 2019). Smale et al. (2019) identified the North Sea as an area where many species live near the edge of their thermal**

tolerance. MHWs in the North Sea in recent summers (2018-2022) have been associated with a collapse in dominant zooplankton populations, with physiological thermal limits exceeded for some species, indicating a significant impact of MHWs on zooplankton (Semmouri et al., 2023). MHWs are also likely to have an impact on chlorophyll-a concentration (CHL), which is a common indicator of phytoplankton biomass and essential for important biogeochemical processes (e.g., oceanic carbon sequestration and export). CHL in the North Sea is strongly influenced by sea surface temperature (SST), nutrient levels, and light conditions (Desmit et al., 2020). Recently, Alvera-Azcárate et al. (2021) pointed out the dominant role of SST on the timing of the spring bloom in the North Sea. They also observed a phenological shift, with the spring bloom occurring earlier each year, by about one month from 1998 to 2020. Generally, MHWs are associated with a decrease in CHL in the tropics and mid-latitudes, and an increase at high latitudes (Noh et al., 2022). However, the response of CHL to MHWs in the North Sea remains unclear.

In the methodology:

- For the CHL analysis, we first analyze the seasonal variation and spatial trend of CHL over the period (1998–2024). Then, to investigate the potential impact of MHWs and MCSs on the CHL concentration in the North Sea, we redetermined the characteristics of MHWs and MCSs based on the climatological baseline of the period overlapping with the CHL (1998–2024). Subsequently, the CHLA is correlated with the total number of MHW days and MCS days.

Regarding the question about the MHW event in spring 2024: That's a very good question. Thank you for bringing this to our attention. In the revised version, we have added the answer to this question at the end of section 3.4 and emphasized it as follows:

- **These results suggest that, in addition to long-term warming, which has contributed to the increase in the MHW occurrence in recent decades (Fig. 7), natural variability and atmospheric circulation may also play a role in modulating (i.e., either amplifying or attenuating) MHW characteristics (Chen and Staneva, 2024; Mohamed et al., 2023). Particularly in shallow water regions such as the North Sea, where atmospheric circulations over the North Atlantic and European region, including the North Sea, are likely to influence SST variability due to the rapid response of shallow water seas to atmospheric forcing (Atkins et al., 2024).**

Regarding the timing of chl-a and SST/MHW changes: To address this point, we reanalyzed the chl-a phenology using the same temporal breakpoint as the SST/MHW analysis. This alignment allows for a more direct comparison and

**shows that the shift in Chl-a timing (e.g., earlier onset of bloom) is more pronounced in the post-2013 period and coincides with an increased frequency and intensity of MHWs. In addition, as mentioned above, we have added a more robust and in-depth analysis of the relationship between the CHL and the total days of MHW and MCS.**

One of the metrics used is MHW frequency, which does not account for the duration of events. An increase in heatwave frequency does not necessarily mean more time under MHW conditions compared to longer, less frequent events. Number of heatwave days per year would be a better metric.

**A: Thank you for this valuable suggestion. We agree with you that MHW frequency alone does not fully capture the temporal extent of MHW conditions. In the revised version, we have added the temporal evaluation and trend of the total number of MHW days (Fig. 7B) as well as their spatial trend (Fig. 8B). In addition, in the previous version we used the cumulative MHW intensity (ºC. days) as the cumulative intensity can simultaneously reflect the frequency, duration, and mean intensity of MHW (Jin and Zhang, 2024), which is a more comprehensive metric for assessing the cumulative thermal stress.**

Please include a description of what constitutes and impacts on "internal variability" which is the other main driver of MHW change along with SST increases.

**A: In the revised version, we have added and highlighted this point as follows:**

- **The long-term trends are most likely dominated by the external anthropogenic forcings, while the internal variability refers to natural fluctuations in the coupled ocean-atmosphere system, including hydrodynamic processes (e.g., currents, mixing, stratification) and atmospheric variability (e.g., wind and pressure patterns). These processes can play a role in SST variability, which can amplify or suppress MHW development on seasonal to interannual time scales. For example, Mohamed et al. (2023) found that the change in atmospheric circulation over the southern North Sea in April 2013 led to an extremely cold event, while in the same month of the following year (2014) it led to an extremely warm event.**

**Detailed comments**

Lines 15-16 there is no analysis shown relating SST to the AMO and EAP (only to MHW intensity PC1 and AMO/EAP) so the "crucial role" statement is not supported here.

**A: In the revised version of the manuscript, we have added and highlighted the correlations between SST with both AMO and EAP, which support this statement.**

- **We also examined the relationship between the PCs and the normalized indices of AMO and EAP (bars and green line in Fig. 6C). The first PC1 showed a significant correlation (r) with both AMO (r = 0.66, p<0.05) and EAP (r = 0.50, p<0.05). In addition, the SST of the North Sea showed significant correlations with the AMO (r = 0.79, p<0.05) and the EAP (r = 0.56, p<0.05).**

Line 16: the doubling of the SST trend is an important result, consider adding the rates in the abstract.
**A: We have added the trend rates in the abstract as follows:**

- **In particular, the SST trend has doubled in the post-2013 period (0.8°C/decade) compared to the pre-2013 period (0.4°C/decade), leading to longer and more frequent MHWs.**

Lines 25-27: please clarify the sentence on chl-a results, However the results were mostly not significant so probably not appropriate for the abstract.
**A: In the revised version, we have rewritten the sentence as follows, based on the new analysis of the correlation between CHL and MHW days.**

- **Finally, we also investigated how the chlorophyll-a concentration responded to the MHW, revealing a decrease in the deep and cold-water regions of the northern North Sea and an increase in the shallow and warm water areas of the southern North Sea.**

Line 47: since the AMO and EAP are used in the current analysis please include brief overviews of their major features and how they impact on SST/MHWs.
**A: In the revised version, we have added and highlighted this point as follows.**

**- These large-scale climate modes are associated with SST and atmospheric variability in the North Atlantic, ranging from interannual (e.g., NAO) to decadal or longer timescales (e.g., AMO and EAP), which can influence the likelihood of MHW in this region (Holbrook et al., 2019).**

Line 57-58: "Therefore, climate change …" does not follow from the previous sentence on the North Sea being a productive fishery. Please rephrase.

**A: In the revised version, we have rephrased the sentence to be as follows.**

- **Climate-related changes and extreme events in this region could have a profound impact on this rich marine ecosystem (Kirby et al., 2007; Smale et al., 2019).**

Lines 64-66: results should not be included in the introduction.
**A: We briefly introduce the previous climate shift here to provide context, while the detailed findings are thoroughly presented in the 'Results' section.**

Line 71: 2004 should be 2024.
**A: Corrected, thanks for catching this typo mistake.**

Line 75: Dataset -> Datasets.
**A: Corrected.**

Line 83: some of these products are not used in the analysis – remove those that aren't used from the list (eg latent heat etc).
**A: In the revised version, we removed the unused data.**

Line 110-112: sentence ending "to detect MHWs" does not make sense. Please clarify.
**A: We have removed this end.**

Lines134-140: definition of PR is not clear: how does the threshold for P1 change each year? Please clarify.

**A: In the revised version, we have reworded the entire paragraph as follows to simplify the definition.**

- **The PR is estimated as P1/P0, where P1 is the probability of MHW days in a specific year, defined as the total number of MHW days observed in that year divided by the total number of days in that year. P0 is the probability of MHW days during the entire study period (1982–2024), defined as the number of MHW days observed in all years divided by the total number of days in all years (43 years\*365.25 days=15706 days). Thus, PR represents the relative strength of the MHW each year compared to the entire study period.**

Line 139 and line 142: what is the MHW "change" defined relative to?
**A: Corrected>>> change in the MHW days.**

Line 151: should be "top left".
**A: Corrected.**

Line 157: should be "top right".
**A: Corrected.**

Line 160: please define the abbreviation SSTA and state how it is calculated.

**A: We have defined the abbreviation of SSTA and stated how it was calculated in the methodology section as follows.**

- **The daily SSTA was calculated by subtracting the long-term average SST for a given day (i.e., the daily climatology) from the observed SST of the same day; the monthly and annual SSTA were then calculated using the daily SSTA.**

Line 160: in addition to the SST increase (0.8 deg C) please add the pre- and post-2013 mean temperatures.

**A: In the revised version, we have added the mean values of the SST in the pre- and post-period as follows:**

**The annual SST has increased significantly by around 0.8°C in recent years (2013–2024: post-2013, hereafter) compared to the previous period (1982–2012: pre-2013, hereafter), with an average SST of 10.67°C and 11.46°C in the pre- and post-2013 periods, respectively.**

Not sure that figure 1C is needed – figure 2 shows a clearer timeseries of SSTA and better supports the discussion (lines 172-174).

**A: We think that Figure 1C is important and illustrative as it shows the temporal evolution of daily sea surface temperature anomalies.**

Line 185: why use the LOWESS method and how are the weights calculated?
**A: Here, we used LOWESS (a non-parametric regression technique) to confirm the climate shift and accelerated warming after 2013. The weights are calculated using a tricube weighting function. We have included the reference of Cheng et al. (2022) for more details on this technique.**

- **The LOWESS trendline (thick black line in Fig. 2) also confirms the acceleration of SST warming after the crucial point of climate shift in the post-2013 period.**

Figure 3B: green line is not defined in the caption.
**A: We have defined the green line, which refers to the SSTA.**

Line 211: How do the SST increases in this study compare with estimates from literature?
**A: We cannot make a comparison here as this value is the difference between the post- and pre-2013 periods, which is not documented in the literature.**

Line 227: please clarify what are the cumulative trends and what do they signify.
**A: The cumulative trend is the annual trend multiplied by the total number of years. However, we have removed this sentence to avoid confusion with the cumulative MHW. Especially, the trend of SST was illustrated in the previous part.**

Lines 229-230: p>0.05 means that the correlations are not significant.
**A: Thank you for spotting this error; the p-value is less than 0.05. We have corrected this in the revised version.**

Line 232: brackets not needed (from 1.6 to 9.6).
**A: Corrected.**

Line 260: increase SSTA does not necessarily imply "an excessive trend in MHW": MHWs are sustained increases (longer than 5 days) in temperature, please be careful of that point.
**A: Corrected>>> leading to an increase in the MHW occurrence.**

Line 312: use "maximum positive variability" instead of "opposite maximum variability", which doesn't make sense.
**A: Corrected.**

Line 314: there is no figure 6D.
**A: Corrected to be Figure 6C.**

Line 315: please clarify the statement "which corroborates a negative trend in variability intensity".
**A: Removed.**

Line 351: typo in 0.1.98.
**A: Corrected to be ± 1.98 (°C. days)/decade.**

Line 367: please elaborate on how the results suggest that "SST variability and thus MHW in the North Sea are largely influenced by atmospheric rather than oceanic forcing".
**A: We have removed the whole section based on the reviewer # 1 suggestion.**

Line 398 and elsewhere: end date of the spring 2024 event is given as July when it should be June.
**A: Corrected.**

Line 417: please elaborate or delete this statement on "atmospheric overheating".
**A: In the revised version, we rewrote the sentence as follows to define what we mean by overheating.**

- **This suggests that atmospheric overheating (i.e., the increase in air temperatures compared to their climatological values at this time of year) and thus weather conditions could best explain this MHW event.**

Figure 11: Is the caption correct "all anomalies were calculated by subtracting the daily climatological SST"? The text on the subfigures is too small and blurry, labels A-C are difficult to read. The figure needs more explanation, e.g. how are the anomalies calculated – is the seasonal signal removed? The spring 2024 event was obviously severe, it is possible to say why it was so much more severe than others?

**A: Thank you for recognising this mistake. We corrected to be: All anomalies were calculated relative to the daily climatological baseline (1982–2024). The resolution of the figure and labels is very clear in the Word file, perhaps only in the PDF file. If needed, we will provide the production section with a clearer copy (400 dpi). We have also described the calculation of the anomalies in more detail in the methodology section. This event was attributed to the anomalous anticyclonic circulation in addition to long-term warming.**

Line 466: "smaller trend" denotes a comparison, but compared to what?
**A: Corrected>>> small**
Line 472: there is no blue line in figure 12.
**A: Corrected to be green line.**

Lines 473-475: figure 4C does not support this statement – it shows differences before and after 2013, not 2009.
**A: We have rewritten the whole sentence and illustrated this statement based on 2013, not 2009**.

Figure 12: yellow text is difficult to read and should be replaced. Please specify what the dots signify in B and C.
**A: We have replaced the yellow text with the blue text. We also clarify that the dots indicate an insignificant trend or correlation (in Figure 13 in the revised MS).**

Line 501: how is the chl-a anomaly calculated?
**A: The CHL anomaly is estimated in the same way as the SSTA. We have added and highlighted this in the methodology section.**

Figure 13D: white contour is not visible.
**A: We have removed the figure and replaced it with another figure.**

Lines 550-551: the response of chl-a to MHWs and MCSs was not so clear-cut as a north-south split, please clarify.
**A: That's right, in the revised version, we rewrote it based on the new analysis of the correlation between CHLA and the total numbers of MHW and MCS days.**

**We hope these revisions and clarifications address your concerns. Thank you once again for your valuable feedback and constructive comments.**

---

## Author Response (AR2)

**Response to the Editor's comments**

Dear Dr Mohamed,

The two reviewers are satisfied with the revised version of your manuscript, except for a small typo to fix. However I have noted a few points in the third review that we unfortunately misplaced. These are minor but I think that addressing them will improve the quality of the manuscript. Please consider the following points (the line numbers refer to the newest submission, the one with tracked changes):

- A: Once again, we would like to thank you and all the anonymous reviewers for their willingness to improve the quality of our manuscript. We have corrected this typo to be 2013 instead of 2023. In the following, we address all comments/clarifications suggested by the third reviewer point by point. The responses are shown in bold text.
- Lines 135-138, reviewer 3 said: "The method applied is not the "shifting baseline" presented by Amaya et al. (2023). The shifting baseline involves a redefinition of the climatology over shifting 30-year periods. Please refer to the paper by Smith et al. 2025 (https://www.sciencedirect.com/science/article/pii/S0079661124002106) for a more comprehensive discussion of MHW definitions in the presence of a trend, and their implications."
- A: Although most of the previous publications refer to this approach as "shifting baseline", including the proposed article by Smith et al. 2025 (please see the note at the end of the definitions of "detrended baseline" and "shifting baseline" in Table 1 in their article), in the revised version of our manuscript, we have used the term "detrended baseline" instead of "shifting baseline" to distinguish between the two cases, as suggested by Smith et al. 2025.
- Line 178, reviewer 3 notes: "Why is the approach applied twice? This sentence seems incomplete.". Indeed the sentence seems incomplete to me as well, maybe it is just a question of style.
- A: That's right, the word twice could be confusing, so we rewrote the whole sentence as below, especially since we already mentioned in the same paragraph that we reapplied the test again for the second period (2001 to 2024) to investigate whether a new regime shift has recently occurred in the North Sea SST.
- The result of this test, based on the entire study period (1982–2024), reveals that a significant change point in SST occurred between 1996 and 2001 (Fig. 1A, top left panel).
- lines 320-324 (beginning of section 3.3), reviewer 3 noted: "We already know that the trend is likely nonlinear given the two detected climate shifts and the trend intensification after 2013. The rationale for and implications of considering a linear trend should be discussed." I got the same impression as the reviewer when I read the paragraph. Maybe it is something you would want to address?

A: This is one of the questions that still has a lot of debate among scientists, for example, the above-mentioned article by Smith et al. (2025) demonstrates that "Often, assumptions about the nature of warming (e.g., linear, or higher-order trends) are made, with the risk that higher-order trends may inadvertently remove important low-frequency natural variability along with the anthropogenic warming signal". In our manuscript, the decision to include a linear trend analysis in this section was motivated by the need to provide a baseline quantification of long-term changes over the full study period. Linear trends are commonly used in most previous studies as a first-order approximation to facilitate comparison across regions and time frames. In the revised MS version (lines 377-383), we have added and highlighted the justification and limitations of using the linear approach.

- Another remark for reviewer 3: "Figure 6. The AMO clearly shows a positive trend encompassing the whole study period. How much does the AMO evolution contribute to the estimated linear trend? It seems that associating the linear trend to climate change may be incorrect as that trend may also include internal variability. This aspect should be carefully discussed." I find this remark relevant. Maybe a caveat should be added to the discussion in lines 369-374? Or elsewhere in the manuscript?

A: That's a good point. However, in this figure, we have not removed the trend before the EOF analysis of cumulative MHW intensity, as the aim here is to analyze the total variability (i.e., without removing the trend) and not to separate the role of trend and internal variability (this point has been addressed in the same section based on the TAR). Therefore, we used the non-detrended AMO index (i.e., including internal variability and trend). This figure suggests that the observed SST regime shift and associated MHW are attributed to both the long-term warming trend and the shift to the warm phase of the AMO mode. In the revised version, we emphasized in the caption of this figure that the EOF was applied based on the original SST.

I hope you will find these comments relevant; taking them into account should not take much of your time. The manuscript will not be sent out to the reviewers again.

A: Thank you again for your time. We really appreciate it. All comments are relevant, and we consider all of them as mentioned above.

We also consider the editorial board's comment on the use of the color blindness simulator scheme for Figures 4, 6, 7, 10, and 12.

Yours

sincerely,

Anne Marie Treguier